

# In vitro effect of resin infiltrant on resistance of sound enamel surfaces in permanent teeth to demineralization

Meng Li, Zhengyan Yang, Yajing Huang, Yueheng Li and Zhi Zhou

Stomatological Hospital of Chongqing Medical University, Chongqing, China
Chongqing Key Laboratory of Oral Diseases and Biomedical Sciences, Chongqing, China
Chongqing Municipal Key Laboratory of Oral Biomedical Engineering of Higher Education, Chongqing, China

Corresponding authors
Yueheng Li,
YF@hospital.cqmu.edu.cn
Zhi Zhou, 2727533620@qq.com

## ABSTRACT

**Objective**. To investigate the effect of resin infiltrant on resistance of sound permanent enamel surfaces to demineralization.

**Method**. Eighty healthy premolars were sectioned to obtain enamel blocks from the buccal surface. Specimens with baseline surface microhardness values of 320–370 were selected. The experimental group were treated with resin infiltrant, while the control group was not. Specimens from each group were artificially demineralized and the surface microhardness values were measured again. Confocal laser scanning microscopy was used to measure the depth of demineralization and detect the penetration ability of the resin infiltrant. The specimens were subjected to a simulated toothbrushing abrasion test. Scanning electron microscopy was used to observe changes in the surface morphology of specimens after each of these procedures.

**Results**. No significant differences between the experimental and control groups were observed in the baseline microhardness values or in the experimental group after resin infiltration compared with the baseline conditions. After artificial demineralization, the microhardness value in the control group was significantly lower than that in the experimental group (266.0 ($\pm$34.5) compared with 304.0 ($\pm$13.0), $P = 0.017$). Confocal laser scanning microscopy results showed that the demineralization depth in the control group was significantly deeper than that in the experimental group (97.9 ($\pm$22.8) $\mu$m vs. 50.4 ($\pm$14.3) $\mu$m, $P < 0.001$), and that resin infiltrant completely penetrated the acid-etched demineralized area of the tooth enamel with a mean penetration depth of 31.6 ($\pm$9.0) $\mu$m. Scanning electron microscopy showed that the surface morphology was more uniform and smoother after simulated toothbrushing. The enamel surface structure was more severely destroyed in the control group after artificial demineralization compared with that of the experimental group.

**Conclusion**. Resin infiltrant can completely penetrate an acid-etched demineralized enamel area and improve resistance of sound enamel surfaces to demineralization. Our findings provide an experimental basis for preventive application of resin infiltrant to sound enamel surfaces to protect tooth enamel against demineralization.

## INTRODUCTION

Resin infiltrant is a low-viscosity (*Ammari et al., 2014*; *Oliveira et al., 2020*), light-cured resin with high penetration ability (*Paris, Meyer-Lueckel & Kielbassa, 2007*; *Kielbassa, Muller & Gernhardt, 2009*). The basic principle of resin infiltration is to penetrate and occlude the porous volume of subsurface lesions by capillary forces, thereby partially or completely replacing the missing minerals, enveloping the hydroxyapatite crystals, micromechanically interlocking the remaining enamel prisms. This method effectively constructs a covalently bonded three-dimensional polymer framework (*Kielbassa, Muller & Gernhardt, 2009*; *Kashbour et al., 2020*) and occluding diffusion pathways for cariogenic acids and dissolved minerals to arrest proximal subsurface lesion progress (*Meyer-Lueckel et al., 2011*; *Askar et al., 2018*). Previous studies on resin infiltration have mainly focused on its effect on non-cavitated proximal lesions (*Araújo et al., 2015*; *Arthur & Zenkner, 2018*), enamel white-spot lesions (*Markowitz & Carey, 2018*; *Silva et al., 2018*; *Youssef et al., 2020*), fluorosis (*Sekundo & Frese, 2020*), and dentin hypersensitivity (*Liu et al., 2015*). The technique is effective in preventing the progression of initial caries in primary and permanent teeth (*Faghihian et al., 2019*), and treatment of early dental caries with resin infiltrant achieved excellent clinical results (*Lasfargues et al., 2013*; *Schwendicke et al., 2014*; *Faghihian et al., 2019*; *Youssef et al., 2020*). Resin infiltration is a non-invasive dental treatment option that complements the concept of minimum intervention dentistry (*Lasfargues et al., 2013*), and its use is closing the gap between application of oral hygiene and minimally invasive dentistry (*Kielbassa, Muller & Gernhardt, 2009*).

Dental caries is the most common chronic oral disease in children and adults worldwide, and is the main cause of defects in dental hard tissue and oral pain, which can seriously affect people's quality of life. In recent decades, the management of dental caries has shifted from drilling and filling to prevention, control, and minimally invasive operative repair in order to preserve more dental tissue (*Lasfargues et al., 2013*). Fluoride has been widely used for the prevention of dental caries since the mid-20th century (*Oh et al., 2017*). Dental sealants were introduced in the 1960s to help prevent dental caries, mainly in the pits and fissures of occlusal tooth surfaces. Sealants act to prevent bacterial growth that can lead to dental decay (*Faghihian et al., 2019*). Research has shown that application of fluoride varnish or resin-based fissure sealants to first permanent molars helps prevent occlusal caries (*Kashbour et al., 2020*). In general, these preventive measures will be affected by people's compliance, oral hygiene habits, etc. Therefore, it is the unremitting pursuit of the majority of medical workers to continuously find more ways to prevent dental caries.

At present, there is no research on the use of resin infiltrant penetrating agent to prevent oral health. Therefore, in the present study, we explored the possibility of using resin infiltrant for primary prevention of dental caries. We used resin infiltrant to treat sound enamel surfaces and performed artificial demineralization, then measured the changes in surface hardness of the enamel and assessed the demineralization depth and degree of penetration. At the same time observe the changes in the surface morphology of samples during the experiment. The effect of resin infiltrant on resistance of tooth enamel to acid erosion and demineralization was explored.

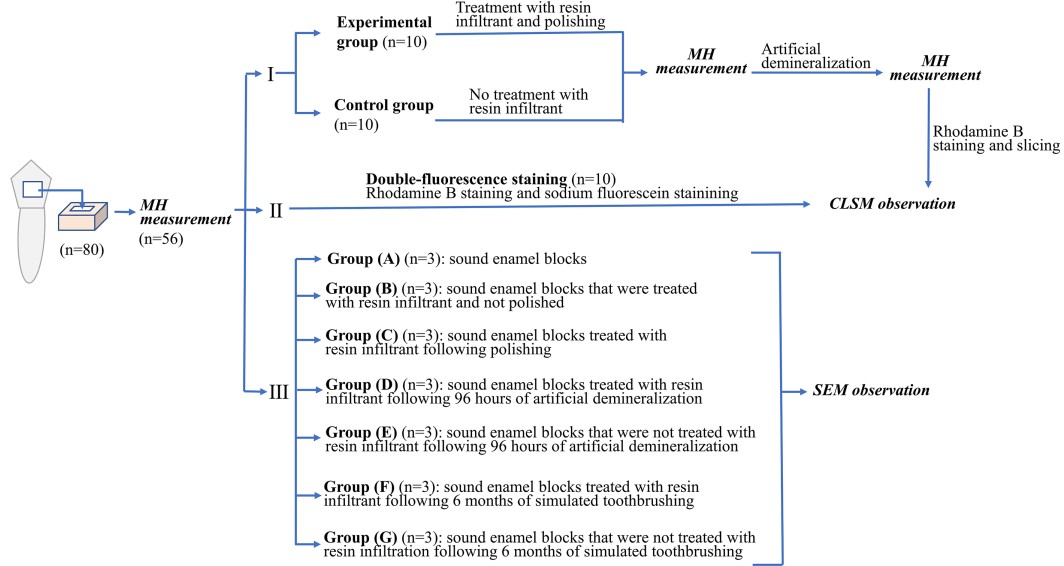

**Figure 1  Experimental flow chart related to the experimental procedures in this study.**

The null hypothesis tested was that the ability of sound enamel surfaces to resist to acid erosion and demineralization will not differ from resin infiltrated enamel previously acid-etched and demineralized.

## MATERIALS & METHODS

The Stomatological Hospital of Chongqing Medical University granted ethical approval to carry out the study within its facilities and participant's written consent has been obtained before tooth extraction. The ethical approval date is May 15, 2018, and the number is CQHS-REC-2018(LS No. 22).

Experimental flow chart related to the experimental procedures in this study was shown in Fig. 1.

### Tooth selection and sample preparation

A total of eighty premolars from patients who underwent orthodontic treatment involving extraction of premolars in the Department of Maxillofacial Surgery of Stomatological obtained from all subjects prior to sample collection. After removal of roots and soft tissue, the premolars were observed under a fully automatic fluorescence stereomicroscope (Leica M205FA; Leica Microsystems, Baden-wuerttemberg, Germany) to ensure that they were intact and contained no cracks or white spots. The teeth were stored in 0.1% thymol solution (Solarbio; Solarbio, Beijing, China) until use, which was within one month of extraction. Enamel blocks (four mm × 4 mm × 2 mm) were obtained from the buccal surfaces of the teeth using a hard-tissue cutting machine (EXAKT 300CP; EXAKT, Hamburg, Germany). The enamel surface was lightly polished in sequence with #400, #800, #1200, #2500, and #5000 silicon carbide abrasive sandpaper ((MATADOR; Eastern supplier, Remscheid, Germany) under running water to create a flat surface, leaving an

exposed window of three mm × 3 mm in the center of the enamel surface (*Gurdogan, Ozdemir-Ozenen & Sandalli, 2017*), while the remaining part was coated with two layers of acid-resistant nail varnish (Maybelline; Maybelline New York, NY, USA). The specimens were then embedded in a denture base resin (FEIYING; Yingpai Dental Materials, Henan, China) and placed in a mold to form cubes of one $cm^3$ (one cm × 1 cm × 1 cm), with the buccal enamel surfaces exposed.

## Microhardness measurement

Surface microhardness (MH) values of the specimens were measured using a Vickers microhardness tester (HV-1000A; Wowei Technology, Beijing, China) with a 200 g load applied for 15 s (*Yazkan & Ermis, 2018*). MH of each specimen was measured at the central, left upper, left lower, right upper, and right lower regions of the exposed enamel window, from which the mean surface MH value of each specimen was calculated. Specimens with a baseline MH value of 320–370 and with the error within 20 between the five measurement points were selected. Finally, fifty-six enamel blocks were selected, and twenty-four enamel blocks were excluded. MH of each specimen was measured before treatment, after resin infiltration and polishing, and after artificial demineralization.

## Resin infiltration and polishing

Randomly select twenty enamel blocks from the included fifty-six enamel blocks and evenly divided into experimental and control groups. In the experimental group, the enamel blocks were treated with resin infiltrant (Icon; DMG Chemisch-Pharmazeutische Fabrik, Hamburg, Germany) (which contained Icon-Etch, Icon-Dry and Icon-Infiltrant). Etching with Icon-Etch for 30 s was followed by 30 s of rinsing and 20 s of drying. Icon-Dry was then applied for 30 s, followed by air-drying to maintain the dryness. When applying Icon-Dry, discoloration in the opaque white areas should be markedly reduced, otherwise the etching and drying processes must be repeated (a maximum of two times). Icon-infiltrant was then applied for 3 min, followed by removal of excess resin and 40 s of light curing. Icon-infiltrant was then reapplied and allowed to soak for 1 min, followed by removal of excess resin and 40 s of light curing. Then, the surfaces of specimens were polished using a Rainbow polishing system (SHOFU; Shofu, Kyoto, Japan) using the black, purple, green, and red polishing discs in sequence for 15 s each at 10,000–12,000 rpm and a pressure of 0.3–0.6 N. The polished specimens are re-applied with double-layer of acid-resistant nail varnish. Specimens in the control group were not treated with resin infiltrant, but prepared as enamel blocks.

## Artificial demineralization

An artificial demineralization solution was prepared (*Zhao & Gao, 2014*) consisting of 2.2 mmol/l $Ca(NO_3)_2$, 2.2 mmol/l $KH_2PO_4$, 50 mmol/l $CH_3COOH$, 5.0 mmol/l $NaN_3$, and 0.01 mmol/l NaF. The final pH was adjusted to 4.5 with NaOH. Specimens were immersed in a beaker (SHUNIU; Shubo, Sichuan, China) containing the demineralizing solution and artificially demineralized for 96 h. The proportion of demineralizing solution per area of exposed enamel window was 2 ml/$mm^2$ (*Rocha et al., 2011*). The beaker was then sealed and placed in a constant-temperature shaker (BIOBASE; Biobase biological, Shandong,

China) at 37 °C (57 rpm/min). The pH value of the demineralizing solution was checked daily and maintained at pH 4.5. The specimens were artificially demineralized for 96 h (*Huang & Li, 2012*).

## Staining and confocal laser scanning microscopy observation

After artificial demineralization, the experimental and control groups were stained with 0.1% Rhodamine B (Solarbio; Solarbio, Beijing, China) solution for 12 h, allowing the red fluorescent dye to fully mark the pores of the demineralized enamel. Specimens were cut longitudinally at the exposed enamel window area into 1.0 mm-thick slices using a hard-tissue cutting machine. The surfaces were then ground into 300 $\mu$m-thick slices with sandpaper under running water (*Huang & Li, 2012*).

### Double-fluorescence staining

In addition, ten enamel blocks were randomly selected from the included fifty-six enamel blocks for double-fluorescence staining. Etched with Icon-Etch, stained with 0.1% Rhodamine B ethanol solution for 12 h, then treated with resin infiltrant. After 300 $\mu$m-thick thin slices were sectioned, they were incubated in 30% hydrogen peroxide (Sanpu; Xi'an Sanpu Chemical Reagent, Xi'an, China) and placed in 37 °C for 12 h to bleach the Rhodamine B solution that had not been enclosed by the resin infiltrant. The slices were then immersed in a 50% ethanol solution containing 100 umol/l sodium fluorescein (Solarbio; Solarbio, Beijing, China) for 3 min and washed with deionized water for 10 s.

The specimens were subsequently observed by confocal laser scanning microscopy (CLSM) (Leica TCS SP8; Leica Microsystems, Baden-wuerttemberg, Germany) under 400× magnification. The excitation and emission wavelengths of Rhodamine B were 568 nm and 590 nm, respectively; those of sodium fluorescein were 488 nm and 525 nm, respectively. ImageJ (ImageJ 1.8.0 for Microsoft; National Institutes of Health, Bethesda, USA) software was used to measure the demineralization and penetration depths. The mean value of five measurements was calculated for each specimen.

## Simulated toothbrushing

In addition, the specimens were brushed using an electric toothbrush (Oral-B P3000; Braun Oral-B/Procter & Gamble, Schwalbach am Taunus, Germany). Toothpaste slurry was made by mixing 5 g toothpaste (Cold Acid Ling; Dengkang Oral Care Products, Chongqing, China) with 15 ml artificial saliva (Solarbio; Solarbio, Beijing, China) (main ingredients: deionized water, NaCl, KCl, $Na_2SO_4$, $NH_4Cl$, $CaCl_2 \cdot 2H_2O$, $NaH_2PO_4 \cdot 2H_2O$, $CN_2H_4O$, NaF; pH: 6.5–7.0) using an electromagnetic stirrer until a homogeneous suspension (slurry) resulted (*Kielbassa et al., 2005*). The electric toothbrush could oscillate or rotate at a frequency of 7600 rpm and provided a brushing force of 2 N (standardized vertical loading force) (*Zhao et al., 2017*; *Lee et al., 2019*). The simulated brushing time was calculated on the basis of a brushing time of 120 s twice a day of 28 teeth in the mouth. A tooth has multiple surfaces to be brushed, so the maximum contact time per tooth surface is reported to be 5 s per day; therefore, the simulated brushing time of 15.2 min for the surfaces of the specimens was evaluated to be equivalent to 1.5 years of tooth brushing.

During the period of simulated toothbrushing, the specimens were placed in artificial saliva overnight (*Lee et al., 2019*).

## Scanning electron microscopy observation

Twenty-one enamel blocks were selected from the included fifty-six enamel blocks and divide them into seven groups for scanning electron microscopy (SEM) (ZEISS; ZEISS Auriga FIB Crossbeam System, Baden-Wurttemberg, Germany). (The remaining excess enamel block is used to supplement the loss of enamel block during the experiment.) Group (A) was sound enamel blocks that did not treated with resin infiltrant. Group (B) was sound enamel blocks that were treated with resin infiltrant and not polished. Group (C) was sound enamel blocks treated with resin infiltrant following polishing. Group (D) was sound enamel blocks treated with resin infiltrant following 96 h of artificial demineralization. Group (E) was sound enamel blocks that were not treated with resin infiltrant following 96 h of artificial demineralization. Group (F) was sound enamel blocks treated with resin infiltrant following 6 months of simulated toothbrushing. Group (G) was sound enamel blocks that were not treated with resin infiltration following 6 months of simulated toothbrushing. Then, these specimens were ultrasonically cleaned for 5 min, dried to the conventional critical point, and coated using an ion spatter. Surface morphological changes of the specimens were observed under SEM at a magnification of 3000×.

## Statistical analysis

All statistical analyses were performed by SPSS 20.0 statistical software (SPSS 20.0 for Windows; IBM Analytics, Armonk, NY, USA). Data were expressed as means ±standard deviation (SD). Assumption of normal distribution was checked using Kolmogorov–Smirnov and Shapiro–Wilks tests, and analyzed using one-way analysis of variance (ANOVA). A $t$-test was performed to compare the difference between the two groups. Significance levels of $\alpha = 0.05$ indicated significant differences.

# RESULTS

## Surface microhardness values of specimens in two groups

As shown in Fig. 2, there was no statistically significant difference in the baseline MH values between the experimental and control groups (344.8 (±6.0) *vs.* 349.0 (±9.2), $P = 0.240$), indicating that there was no significant difference in the degree of initial enamel mineralization between specimens in the two groups. After resin infiltration, the surface MH value of samples in the experimental group was 346.8 (±9.7), with no significant difference ($P = 0.250$) when compared with the baseline, indicating that treatment of sound enamel surfaces with resin infiltrant did not affect the degree of mineralization. After 96 h of artificial demineralization of specimens in the two groups, surface MH of specimens in the control group was significantly decreased compared with that in the experimental group (266.0 (±34.5) *vs.* 304.0 (±13.0), $P = 0.017$), indicating that the degree of demineralization was higher in the experimental group.

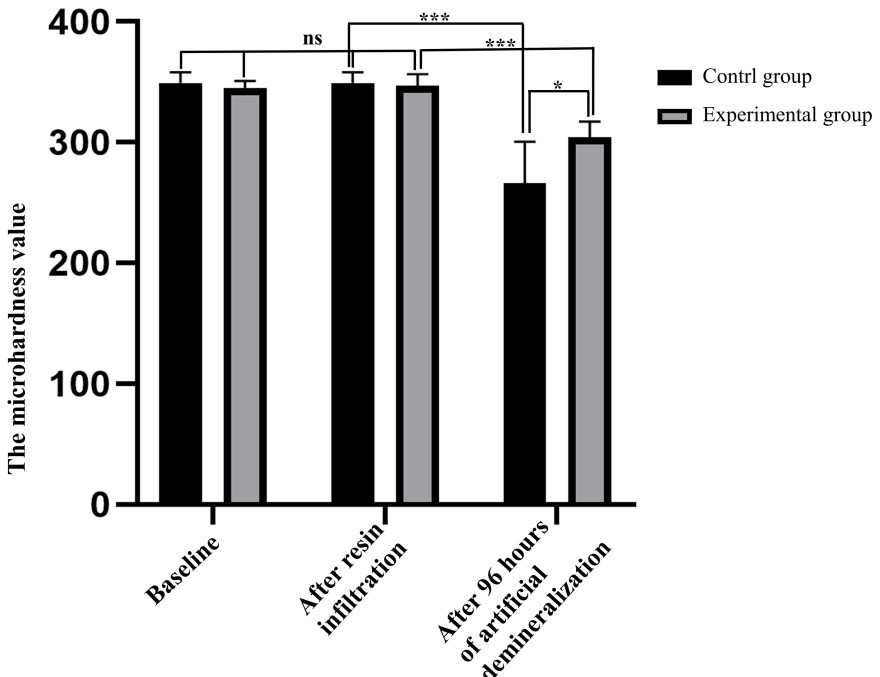

**Figure 2** **Comparison of surface microhardness values of specimens between control and experimental groups at each time point ($n = 10$, mean $\pm$ SD).** *$P < 0.05$, ***$P < 0.001$, ns = no significant difference.

## Confocal laser scanning microscopy results

Red areas in the CLSM images represent the depth of the demineralization in specimens stained with Rhodamine B and the acid-etched enamel areas sealed by the resin infiltrant; green areas represent the demineralized microporous areas that were not sealed by resin infiltrant; black areas represent air and normal hard tissues that were not stained. The results showed that the demineralization depth in resin-infiltrated specimens (experimental group) was significantly shallower than that in non-resin-infiltrated specimens (control group) (50.4 ($\pm$14.3) $\mu$m *vs.* 97.9 ($\pm$22.8) $\mu$m, $P$ <0.001; Figs. 3 and 4). Double-fluorescence staining showed that the resin infiltrant almost completely penetrated the etched demineralized enamel areas, with a mean penetration depth of 31.6 ($\pm$9.0) $\mu$m (Fig. 5).

## Scanning electron microscopy results

Changes in enamel surface morphology observed by SEM showed that the surfaces of untreated enamel blocks only had visible scratches (Fig. 6A). After the samples were treated with resin infiltration and not polished, the enamel surfaces showed a dense, rough, fish-scale shaped, enamel prism-like structure (Fig. 6B). After polishing, changes in the surface structure were not obvious and the enamel surfaces became more uniform (Fig. 6C). After 96 h of artificial demineralization, the surfaces of the untreated enamel blocks exhibited obvious pit-like structures (Fig. 6E) and the enamel prisms of untreated enamel blocks were more severely damaged than those of resin-infiltrated enamel (Fig. 6D). After

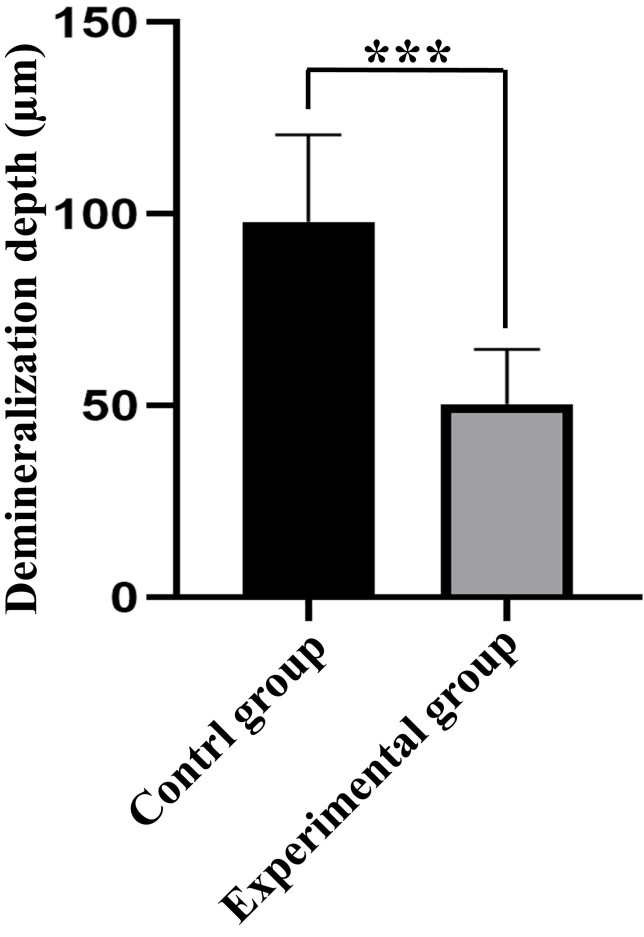

**Figure 3 Comparison of demineralization depth of specimens between control and experimental groups (*n* = 10, mean ± SD).** ***P < 0.001.

the simulated toothbrushing abrasion test, the resin-infiltrated enamel blocks showed smooth surfaces with a dense and uniform fish-scale-like structure (Fig. 6F), while the surfaces of the untreated enamel blocks had no obvious special structure and were smooth (Fig. 6G).

## DISCUSSION

The main goals of modern dentistry are early prevention, early detection, and timely treatment of early lesions by promoting remineralization. Resin infiltration is a minimally invasive approach for treating early enamel caries without mechanical destruction of the enamel structure (*Kielbassa, Muller & Gernhardt, 2009*; *Skucha-Nowak, 2015*). To further explore the possibility of using resin infiltrant in the primary prevention of dental caries, in this study, sound permanent enamel surfaces were treated with resin infiltrant followed by further experimental procedures, such as artificial demineralization, simulated toothbrushing, and fluorescent staining to observe changes in the degree of

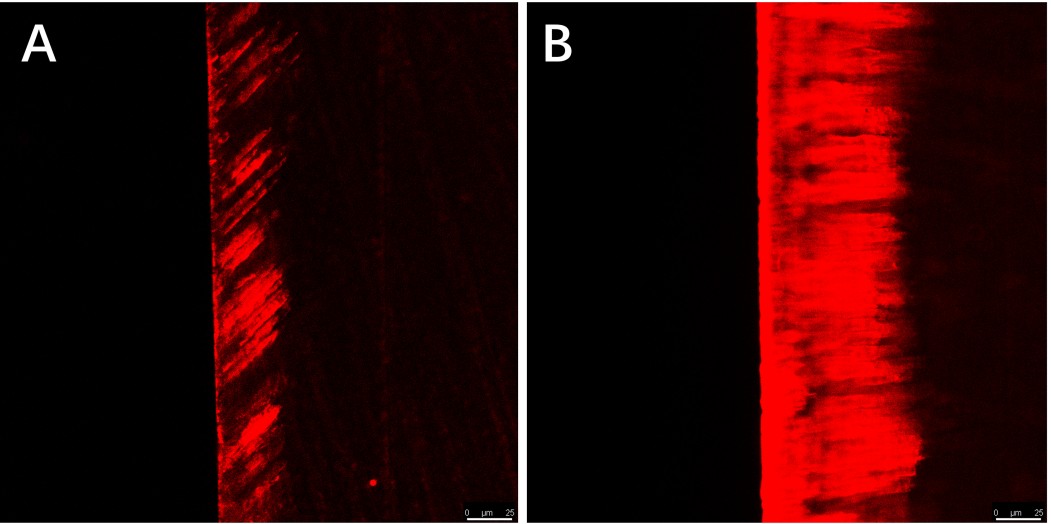

**Figure 4  CLSM images of the specimens after rhodamine B staining, the width of the red area reflects the demineralization depth of enamel.** (A) Staining results of the experimental group specimens after artificial demineralization 96 h. (B) Staining results of the control group specimens after artificial demineralization 96 h. Scale bar represents 25 μm.

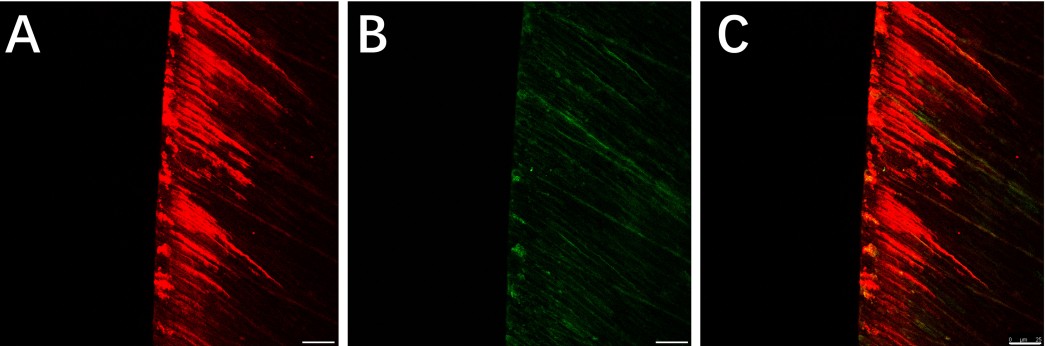

**Figure 5  CLSM images of the specimens after double-fluorescence staining.** (A) Rhodamine B staining results of the specimens after etched with Icon-Etch; red area represents the depth of the acid etched demineralized enamel areas sealed by resin infiltrant. (B) Sodium fluorescein staining results of the specimens after treated with resin infiltrant, sectioned and bleached; green area represents the demineralized microporous areas that are not sealed by resin infiltrant. (C) Merge of (A) with (B); shows that resin infiltrant almost completely penetrated into the etched demineralized enamel areas. Scale bar represents 25 μm.

demineralization and surface morphology of the enamel. The results of this study show that the infiltrating resin can improve the ability of a sound enamel surface to resist acid corrosion and demineralization and completely penetrate the acid-etched demineralized enamel area. Thus, the null hypothesis was rejected.

MH can be used as a parameter to detect changes in the mineral content of dental hard tissue (*Gomez et al., 2008*); MH changes can reflect mineral loss or gain (*Kielbassa et al., 1999*). In our study, results from MH measurement showed that preventive application of

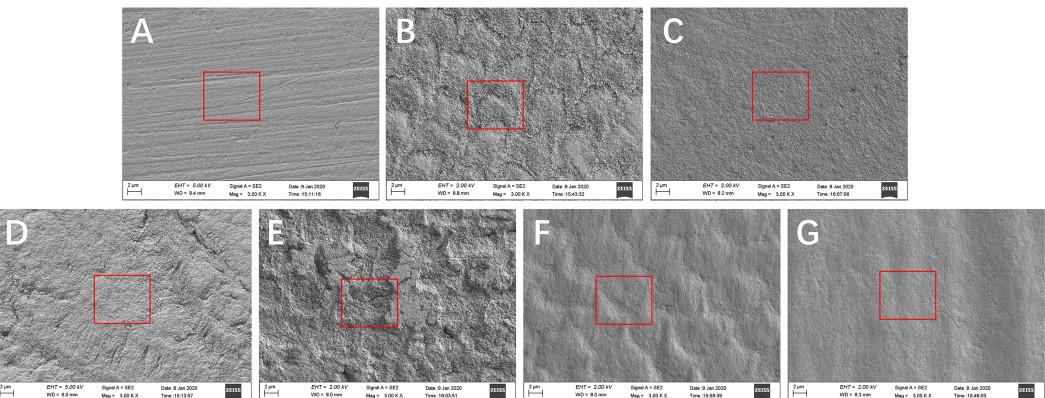

**Figure 6** **SEM resutls of surface morphology.** (A) Sound enamel blocks that did not treated with resin infiltrant. (B) Sound enamel blocks that were treated with resin infiltrant and not polished. (C) Sound enamel blocks treated with resin infiltrant following polishing. (D) Sound enamel blocks treated with resin infiltrant following 96 h of artificial demineralization. (E) Sound enamel blocks that were not treated with resin infiltrant following 96 h of artificial demineralization. (F) Sound enamel blocks treated with resin infiltrant following 6 months of simulated toothbrushing. (G) Sound enamel blocks that were not treated with resin infiltration following 6 months of simulated toothbrushing.

resin infiltrant has no significant on sound enamel surfaces on its hardness. This finding provides a theoretical basis for further experiment.

After artificial demineralization, the degree of demineralization was lower in the experimental group and that resin-infiltrated enamel had strong ability to resist acid erosion and demineralization. The triethylene glycol dimethacrylate (TEGDMA) present in resin infiltrant and hydroxyapatite in enamel constitute a uniform composite, and that the interaction of these crystals improves enamel resistance to acid erosion and slows the loss of mineral content of enamel (*Kielbassa et al., 2020*). It was previously reported that adding ethanol and TEGDMA to Icon resin can significantly reduce the viscosity and contact angle of the material, thereby increasing the permeability coefficient. The high permeability of penetrating resin can reduce morbidity, microleakage and secondary dental caries (*Chen et al., 2019*). However, some studies have shown that the microhardness of infiltrating resin after treatment of early caries cannot be comparable to that of sound enamel (*Neres et al., 2017*). The penetrating surface did not show complete resistance to new cariogenic challenges (*Neres et al., 2017*; *Torres et al., 2012*). In addition, for caries extending into dentin, treatment efficacy of resin infiltration was not significantly different from the non-infiltrated controls (*Liang et al., 2018*). Therefore, we proposed an experimental design for the treatment of sound enamel surface with resin infiltration. And the results show that infiltrating resin can help sound enamel resist artificial demineralization to a certain extent.

CLSM is a high-resolution microscopy technology that has been widely used in study of dental caries and oral microbiology. In this study, results from CLSM observation after Rhodamine B staining showed that the demineralization depth in the experimental group was less than that in the control group, which confirmed the MH results. This
indicates that resin infiltrant can occlude the micropore structure found in enamel and block the passages that bacteria and acid require to cause further dissolution of the enamel structure. The treated enamel had good mechanical stability that prevented dissolution of the enamel surface structure in an acidic environment, making it more resistant to acid and demineralization. Our result is in line with previous findings by *Gurdogan, Ozdemir-Ozenen & Sandalli (2017)* that showed that the MH of resin-infiltrated demineralized enamel increased. Nevertheless, research has shown that resin infiltration is unable to remineralize the demineralized tooth enamel and to prevent further recurrent caries (*Gelani et al., 2014*). In addition, Studies have pointed out that with progressed enamel carious lesions the infiltration frequently will be inhomogeneous and incomplete (*Schneider et al., 2017*). Therefore, the treatment of infiltrating resin for the formed early caries does not guarantee a 100% success rate. So, we guess that the earlier the use of penetrating resin may have a positive effect on tooth enamel. Our findings further confirm the positive effect of resin infiltrant on improving resistance of sound enamel to acid erosion and demineralization, and provide an experimental basis for the preventive use of resin infiltrant on enamel surfaces.

Double fluorescence staining results showed that treatment of sound enamel surfaces with Icon-Etch resulted in slight demineralization of the enamel surface layer, with a mean demineralization depth of 31.6 ($\pm$9.0) $\mu$m. This result is consistent with findings from previous studies investigating the effect of etching gel on enamel surfaces (*Meyer-Lueckel, Paris & Kielbassa, 2007*; *Paris, Dörfer & Meyer-Lueckel, 2010*; *Neuhaus et al., 2013*; *Arnold et al., 2015*). Acid etching can increase the surface roughness of the enamel, create microporosity in the enamel, and ethanol dehydration can increase the penetration ability of low-viscosity resin (*Kielbassa et al., 2005*; *Ulrich et al., 2015*; *Yoo et al., 2019*; *Youssef et al., 2020*). In the present study, staining results showed that resin infiltrant effectively sealed the micropores in sound enamel formed by an etching gel. Previous studies investigating the effect of resin infiltrant on caries lesions by *Mandava et al. (2017)* and *Liu et al. (2012)* reported much deeper penetration depths of resin infiltrant than that produced only by etchant on the enamel surfaces. This result also suggests that, for slightly superficial damage to an enamel surface caused by acid etchant, resin infiltrant can completely penetrate demineralized areas.

SEM is based on the interaction of electrons with substances and provides sample images in three dimensions that can reflect the surface structure of the samples. In this study, SEM results showed that sound enamel surfaces showed a scratched appearance after grinding with sandpaper (Fig. 6A). Treatment of sound enamel surfaces with resin infiltrant did not destroy the surface structure of the enamel, and a dense, uniform, but rough, fish-scale-like structure was observed on the enamel surfaces before polishing (Fig. 6B); after polishing, the enamel surfaces were more uniform and smoother (Fig. 6C). Similar results were obtained by *Arnold, Meyer & Naumova (2016)* and *Mueller et al. (2011)*. Applying resin infiltrant to enamel caries lesions can provide and maintain enamel surfaces. Yazkan's Research proposed that although resin infiltrants are capable of penetrating deeply into the porous enamel lesion, they cannot form a smooth coat on the lesion surfaces (*Yazkan & Ermis, 2018*). Rough enamel surfaces increase the chance of bacterial adhesion (*Gurdogan,*

*Ozdemir-Ozenen & Sandalli, 2017*) and pigmentation (*Arnold, Meyer & Naumova, 2016*), so polishing is indispensable for resin infiltration.

After artificial demineralization, there were a few cracks and shallow pits on the enamel surfaces which treated with resin infiltrant, and changes in the enamel prism structure were not obvious (Fig. 6D); while, obvious pit-like structures were observed on the enamel surfaces which not treated with resin infiltrant and enamel prism structure was markedly destroyed (Fig. 6E). This further confirmed the abovementioned MH and CLSM results. These findings indicated that resin infiltrant exerted a significant protective effect on tooth enamel.

After six months of simulated toothbrushing, the enamel surfaces that were not treated with resin infiltrant showed no obvious or clear structure (Fig. 6G). However, compared with the surface morphology of enamel that received no treatment and was not submitted to simulated toothbrushing (Fig. 6A), the enamel surfaces became smooth and the scratches largely disappeared. This may be because abrasive ingredients in toothpaste can cause long-term friction on tooth surfaces under certain pressure of electric toothbrushes, which is similar to the long-term slow effect of polishing. Enamel surfaces treated with resin infiltrant showed a dense and uniform fish-scale-like structure after simulated toothbrushing (Fig. 6F), which showed a smoother surface structure compared with the surface morphology of enamel treated with resin infiltrant that was not subjected to polishing and simulated toothbrushing (Fig. 6B), as well as a clearer surface structure compared with enamel treated with resin infiltrant that was polished, but not subjected to simulated toothbrushing (Fig. 6C). These results indicated that simulated toothbrushing can enable resin-infiltrated enamel surfaces to exhibit a clear, dense, uniform, smooth fish-scale-like structure. We speculate that a certain period of routine toothbrushing simulation can make the enamel surfaces more uniform, smoother, and show certain stability. This further provides a favorable basis for preventive application of resin infiltrant on sound tooth enamel surfaces.

The protective effect of resin infiltrant on sound tooth enamel provides a new attempt on prevention of dental caries, which may be used in preventing dental caries in people at high caries risk, such as patients who will undergo orthodontic treatment. According to reports, the incidence of white spot lesions in patients who have not undergone orthodontic treatment is between 11–24% (*Gulec & Goymen, 2019*). The prevalence of white spot lesions after treatment fixed orthodontic appliances is 23%, 50% or even 97% (*Kobbe et al., 2019*). Studies have shown (*Costenoble et al., 2016*; *Gulec & Goymen, 2019*) that during orthodontic bonding, bracket bonding performed immediately or shortly following treatment of demineralized enamel with resin infiltrant did not affect the bonding quality of orthodontic brackets. People at high risk of caries are more likely to suffer from caries than normal people, which not only affects the quality of life, but also affects people's physical and mental health. Our study provides a new possibility for people at high risk of caries.

However, this was an *in vitro* study, so further studies are needed to confirm the feasibility of preventive using resin infiltrant to protect sound tooth enamel against erosion and demineralization. In addition, some studies pointed out resin infiltrant lacks persistent

antibacterial effects and cannot inhibit bacterial growth (*Tawakoli, Attin & Mohn, 2016*). Therefore, some scholars add antibacterial substances, such as silver nanoparticles (AgNP) (*Kielbassa et al., 2020*), quaternary ammonium methacrylate (*Yu et al., 2020*) to provide the effect of the antibacterial resin penetrant. Our research has not made further discussion on this. Therefore, the follow-up direction of our research is diverse and worthy of in-depth consideration.

## CONCLUSIONS

The findings of this study suggest that resin infiltrant can completely penetrate an acid-etched, demineralized enamel area, effectively seal micropores in the enamel, and improve the ability of sound enamel surfaces to resist acid erosion and demineralization, making it difficult for external acids to enter gaps present in the enamel. Resin infiltration can therefore play a role in protection of tooth enamel from erosion by acid and demineralization.

### Funding

This work was supported by the Chongqing Science and Technology Comission (No.CSTC2019jcyj-msxmx0191) and the Chongqing Municipal Health Commission (No.2018MSXM036, No.2018QNXM023, and No.2017ZDXM018). The funders had no role in study design, data collection and analysis, decision to publish, or preparation of the manuscript.

### Grant Disclosures

The following grant information was disclosed by the authors:
Chongqing Science and Technology Comission: CSTC2019jcyj-msxmx0191.
Chongqing Municipal Health Commission (No.): 2018MSXM036, 2018QNXM023, 2017ZDXM018.

### Competing Interests

The authors declare there are no competing interests.

### Author Contributions

- Meng Li, Zhengyan Yang and Yajing Huang performed the experiments, analyzed the data, prepared figures and/or tables, and approved the final draft.
- Yueheng Li and Zhi Zhou conceived and designed the experiments, authored or reviewed drafts of the paper, and approved the final draft.

### Human Ethics

The following information was supplied relating to ethical approvals (i.e., approving body and any reference numbers):

The Stomatological Hospital of Chongqing Medical University granted ethical approval to carry out the study within its facilities (CQHS-REC-2018(LS No. 22).

## Data Availability

Data are available as Supplementary Files.

## Supplemental Information

Supplemental information for this article can be found online at http://dx.doi.org/10.7717/peerj.12008#supplemental-information.

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
