# Peer review of "In vitro effect of resin infiltrant on resistance of sound enamel surfaces in permanent teeth to demineralization"

_PeerJ, doi:10.7717/peerj.12008_

## Round 0.1 · original submission · Major Revisions

There are still a lot of issues remaining; please address them in a thorough revision.

·

Basic reporting

The paper is clear ad straightforward. English needs some minor corrections

The literature review is adequate and appropriate for the paper.

The structure and figures are well presented and relevant to the hypothesis.

Experimental design

The experimental design is stand for this type of studies.

The research question is clearly stated although the originality of the study is not too unique.

The research was developed following the methodology commonly used for these studies. The methods are adequately described.

Validity of the findings

The results are clinically relevant confirming the value of the infiltration technique

The conclusions are based on the obtained results/

Reviewer 2 ·

Basic reporting

- English remains a concern. See, for example, "experimental group were treated (...)". Must read "the experimental group was treated (...)". Authors are encouraged to search help from a native speaker experienced with scientific English (or, alternatively, use a professional editing service).
- A thorough revision of numbers and units would seem mandatory.
- Literature has been partially ignored, and relevant papers are missing. Background must be elaborated more clearly, along with adequate references, thus leading to the aims. Remember that your rationale remains unclear, so elaborate more clearly your innovative impact.
- Article structure would seem acceptable. Raw data shared.

Experimental design

- This is an original research, but not a primary one.
- Rationale remains unclear. Both aims and objectives must be clearly elaborated.
- Investigation would seem acceptable.
- Several details on methodology would seem missing/are considered perfectible.

Validity of the findings

- Outcome would seem sound.
- Underlying data provided, and considered reasonable.
- Conclusions section would seem perfectible/in need in a thorough revision.

Additional comments

Abstract
- "Healthy premolars were selected (...)." How many premolars have been used?
- "Specimens with baseline surface microhardness (MH) values of 310-370 (g/um2) selected, (...)." Please revise for immediate understanding. Use complete sentences, and stick to clear, unambiguous, technically and grammatically correct English throughout you text.
- "experimental group were treated with resin infiltrant" - this was done WITHOUT prior etching? Please clarify.
- Do not be spuriously accurate, and separate numbers and (correct) units. "266.041±34.484g/um2" must read "266.0 (±34.5) g/µm²".
- What is "um"? Do you mean "µm"?
- What means "more severely destroyed"? Be as precise as possible here. Provide complete results, and add exact P values.
- With your conclusions, do not simply repeat your results. Instead, provide a reasonable extension of your outcome.

Introduction
- With every fact stated, please provide adequate references.
- See, for example, "Resin infiltrant is a low-viscosity, light-cured resin with high penetration ability." To guide the future readers into the right perspective on the development, the authors are strongly encouraged to refer to the introductory papers of the resin infiltration concept, see https://pubmed.ncbi.nlm.nih.gov/19639091/, and https://pubmed.ncbi.nlm.nih.gov/17586715/.
- "(...) resin infiltrant has achieved excellent clinical results." References missing. See, for example, https://pubmed.ncbi.nlm.nih.gov/32901234/, and discuss.
- Same with "And resin infiltration is a non-invasive treatment option that fits perfectly with the concept of minimum intervention dentistry (Lasfargues et al., 2013)." Indeed, resin infiltration is "closing the gap between oral hygiene and minimally invasive dentistry", see https://pubmed.ncbi.nlm.nih.gov/19639091/, and refer to that introductory paper.
- Same with "Previous studies on resin infiltration have mainly focused (...)." References missing.
- "In this study, in order to explore (...)." Should read "In the present study, in order to explore (...)."
- Please elaborate BOTH aims AND objectives more clearly. Clarify why this study was necessary, and what significant contributions this paper would add to the literature. Remember that infiltration effects on microhardness, penetration ability of the resin, and protective efficacy, for example, are well known from literature. These aspects would seem confirmative only. Please revise carefully, and remember to convincingly elaborate your primary aims.
- You relevant and meaningful research question must be clearly defined. Please identify the knowledge gap you are trying to fill.
- A clear and indisputable null hypothesis (this must be deducible from the foregoing thoughts) is missing.

Materials and Methods
- Number and date of your ethical approval is missing.
- "A total of 80 premolars from patients who underwent orthodontic treatment involving extraction of premolars in the Department of maxillofacial surgery of Somatological Hospital of Chongqing Medical University were selected." Please combine this information with "Oral informed consent was obtained from all subjects prior to samples collection.". Refer to Helsinki declaration.
- Why did you use 80 extracted premolars, but finally managed to stick to only 20 specimens for MH testing?
- Notwithstanding, please clarify the reasons for your ethical approval. Remember that with extracted teeth an Ethical Vote would not seem necessary, see https://pubmed.ncbi.nlm.nih.gov/31990942/, and discuss.
- With ALL materials (including chemicals) and methodologies, please use general names with your text, followed by (brand name; manufacturer, city, country) in parentheses. Stick to semicolon. Revise thoroughly.
- "(4mm×4mm×2mm)" must read "(4 × 4 × 2 mm³)".
- Same with "3 mm × 3 mm". Must read "3 × 3 mm²". Revise thoroughly throughout your text.
- Do not repeat single aspects, compare "The Stomatological Hospital of Chongqing Medical University granted Ethical approval to carry out the study within its facilities." and "Ethical approval was obtained from the Institutional Review Board of our hospital.".
- "In the experimental group, the enamel blocks were treated with resin infiltrant ((...) contain Icon-Etch (...))". Please see comments given above (Abstract section). Did you Icon-Etch? How long did you use Icon-Etch? Please clarify.
- "The specimens referring to the control group were not treated with the resin infiltrant." OK, but these specimens were neither acid-etched, right? Please clarify.
- Do not use legal terms with your text, since this is not usual with scientific reports. Delete Co., Ltd., and so on.
- Again, please separate numbers and units. "15s" must read "15 s".
- What is "mmol/L"? Do you mean liter? This would be the international symbol, and this should read "mmol/l", right?
- Same with "2ml/mm2". Should read "2 ml/mm²"? Please revise thoroughly.
- Same with "37℃". Must read "37 ℃". The unit is "°C".
- Same with "umol/L". Should read "µmol/l"?
- Please revise for uniform writing, compare "12 hours" and "10s".
- "400 × magnification" must read "400× magnification". Same with "3000×".
- Regarding acidity and abrasivity of your tooth paste, and regarding the time of your brushing procedure, please refer to https://pubmed.ncbi.nlm.nih.gov/16110209/, and discuss these aspects more thoroughly.
- "(...) the specimens were placed in artificial saliva overnight." Please add information on your artificial saliva here.
- All in all, with your revision, please ensure reproducibility of your research by providing complete information.

Results
- "(...) indicating that treatment of sound enamel surfaces with resin infiltrant did not affect the degree of mineralization." First, this is not a result, but an interpretation. Second, this interpretation would seem false. With your methodology using Icon-Etch, there surely will be some mineral loss. Please discuss.
- Same with your "indicating that the demineralization degree". This would not exactly correspond to surface MH. For further reading, you might wish to go to https://pubmed.ncbi.nlm.nih.gov/10217515/. Revise carefully.
- Please provide exact P values here. P<0.05 or P>0.05 would not seem sufficient.
- Again, please revise for your spurious accuracy. See, for example, "47.356±11.687 µm". Revise thoroughly throughout your text (including your tables), and re-edit your Discussion section accordingly.
- Meaning of "vibible"?
- Meaning of "resutls"?
- With your SEM evaluation, this aspect again would seem confirmatory only. Refer to https://pubmed.ncbi.nlm.nih.gov/19639091/, and to https://pubmed.ncbi.nlm.nih.gov/17586715/, and discuss.

Discussion
- Refer to H0 when starting this section.
- "Resin infiltration is a minimally invasive approach for treating early enamel caries (...)." Again, refer to https://pubmed.ncbi.nlm.nih.gov/19639091/. Ignoring such an landmark paper would not seem acceptable.
- The rationale for "treating sound permanent enamel surfaces with resin infiltrant" must be thoroughly discussed.
- Do not use author names with your text, unless for historical reasons. See "(...) Taher et al.(Taher et al., 2011)". Authors previous work will (and must) be acknowledged with your references. With your text, however, please stick to your main thoughts.
- "(...) and ethanol dehydration can increase the penetration ability of low viscosity resin (Yoo et al., 2019)." This would seem right. However, please remember that acids do not remove organic compounds, and de-proteinistaion will be necessary to to improve the infiltration efficacy. See https://pubmed.ncbi.nlm.nih.gov/25483122/, and go to https://pubmed.ncbi.nlm.nih.gov/31990942/ and https://pubmed.ncbi.nlm.nih.gov/32901234/, for further information on organic debris, bacteria, and yeasts having been found in the pores of (demineralized) enamel. Please address that your in vitro study did not control for these aspects, and this must be clearly elucidated, since this is a limitation. Remember to add some paragraphs discussing the strengths and limitations of your study.

Conclusions
- Please exclusively stick to your revised aims (see comments given above). Do not simply repeat your results here (this section is not called "Repetition"). Instead, please provide a reasonable extension of your outcome.
- Many thoughts given with your current conclusions would seem right, not doubt. However, these should be copied and pasted to the Discussion section. Here, again, please stick to your aims.

References
- Please revise for uniform formatting. Style must be: Full List of authors (with initials). Publication year. Full article title. Full title of the Journal (italicized), volume: page extents. DOI: number
- Please revise for complete author names, and re-edit with regard to use of capital letters with reference titles.

All in all, the authors will agree that there are numerous minor and major shortcomings with their submitted draft. No doubt, there are 5 (!) (co-)authors having read and approved this manuscript, right? This reviewer would like to remind ALL (co-)authors to consequently re-edit such a paper prior to submission. This for sure will improve the quality of such a paper, at least with regard to it's content. Further language improvement will be possible by means of an editing service, thus finally facilitating reading. Undoubtedly, these comments do not intend to blame the authors. However, please remember that external reviewers will not (co-) or (ghost-)author your manuscript, so a high initial quality would seem mandatory with any submitted paper.

Again, authors are encouraged to search help from a native speaker experienced with scientific English (or, alternatively, use a professional editing service).

In total, while the authors' findings would seem relevant, this submitted draft is not considered ready to proceed. Major revisions would seem mandatory, is re-review is strongly recommended.

This reviewer does strongly encourage the authors to carefully and thoroughly revise their draft, and this review should help to improve the quality. Good luck, and stay healthy.

Reviewer 3 ·

Basic reporting

The subject is current and very discussed in the literature. The references are current, however, this paper did not insert them in the introduction, this part was poor and need to improve. In text observed some typed errors.
The title (In vitro effect of resin infiltrant on resistance of
sound enamel surfaces in permanent teeth to acid
erosion and demineralization ) described acid erosion, but this work only studied demineralization process simulation of dental caries, not dental erosion.
Figure 1–very confused–not necessary
Figure 2 unnecessary

Experimental design

Materials and methods are very confused. Suggestion inserts a figure with experimental design and draw of each phase.
The polishing of sealant was realized after protection with nail varnish of enamel, this procedure can be compromissed the isolation and affect the demineralization process
Why used 200gf for microhardness analysis?
The surface was analyzed not clearly, it was the enamel infiltrated or around the sealant.

The rhodamine solution stained the polymers and, this way, maybe it can false results.
What moment did you realize the toothbrushing?
As result, the n was 8 and the materials and methods were 10. Why this difference?
Need to improve de discussion and conclusion.

Validity of the findings

Based on the confusion of materials and methods is very difficult the validity of the findings.
Conclusion not clear and not restrict this research.

Additional comments

Need to improve the paper for obtain to the excellence of this journal

·

Basic reporting

Intro & background to show context. Literature well referenced & relevant.
The background is not adequate. Dental caries and dental erosion are completely different lesions, in the title “In vitro effect of resin infiltrant on resistance of sound enamel surfaces in permanent teeth to acid erosion and demineralization”, it seems that the authors objective was to prevent dental erosion, however the introduction was regarding dental caries, which is wrong because the methods used and the images of SEM shows that the simulated challenge (artificial demineralization) resulted in dental erosion lesions. Therefore, the authors might introduce the justification to apply resin infiltrant to prevent dental erosion.
Please read: Lussi A, Carvalho TS. Erosive tooth wear: a multifactorial condition of growing concern and increasing knowledge. Monogr Oral Sci. 2014;25:1-15.
It is not possible to understand the raw data regarding microhardness, there is no identification regarding the groups.

Experimental design

Research question well defined, relevant & meaningful. The research question is confused, because the author described dental erosion and dental caries as similar alterations. Therefore, the research question is not well defined.
Methods described with sufficient detail & information to replicate.
Methods are not well described. Tooth brushing were performed in different specimens or the ones which were demineralized? At first 80 specimens were prepared, how much were used for erosive demineralization? 10 was used for Double-fluorescence staining. It is not clear.

Validity of the findings

In Figure 5(E) Sound enamel blocks that were not treated with resin infiltrant following 96 hours of artificial demineralization, it is clear that the artificial demineralization resulted in enamel loss (erosion) and not in a subsurface caries lesion. And on the other hand, the most adequate method to measure dental erosion is profilometry, to understand the amount of enamel lost, hardness is able to show only the characteristic of the maintained enamel.
Please read: Attin T, Wegehaupt FJ. Methods for assessment of dental erosion. Monogr Oral Sci. 2014;25:123-42.
The discussion is not adequate because authors are discussing regarding dental caries and the lesion resulted from the methods is erosion.
The conclusion might also be rewritten. Be careful with speculation in the conclusion.

---

## Round 0.2 · Major Revisions

Please address the issues raised by the reviewers and resubmit.

Reviewer 2 ·

Basic reporting

• The submission adheres to all PeerJ policies - no, but further revisions should eliminate all shortcomings
• Manuscript is considered clear, unambiguous, and technically correct - no, but further revisions should eliminate all shortcomings
• The article conforms to professional standards of courtesy and expression - yes
• The article should include sufficient introduction and background to demonstrate how the work fits into the broader field of knowledge. Relevant prior literature should be appropriately referenced - no, but further revisions should eliminate all shortcomings
• The structure of the submitted article conforms to an acceptable format of ‘standard sections’ (see our Instructions for Authors for our suggested format) - yes
• Figures are considered relevant to the content of the article, of sufficient resolution, and appropriately described and labeled - yes
• The submission is considered ‘self-contained,’ represents an appropriate ‘unit of publication’, and includes all results relevant to the hypothesis - no, but further revisions should eliminate all shortcomings

Experimental design

• The submission describes original primary research within the Aims & Scope of the Journal - yes
• The submission clearly defines the research question, which must be relevant and meaningful. The knowledge gap being investigated has been identified, and statements have been made as to how the study contributes to filling that gap - partially, and further revisions should eliminate all shortcomings
• The investigation has been conducted rigorously and to a high technical standard - yes
• Methods have been described with sufficient information to be reproducible by another investigator - partially (material and methodology supplier are missing, and will be provided with the revision)
• The research has been conducted in conformity with the prevailing ethical standards in the field - yes

Validity of the findings

• The data are robust, statistically sound, and controlled - yes
• The data on which the conclusions are based have been provided or made available in an acceptable discipline-specific repository - yes
• The conclusions have been appropriately stated, are connected to the original question investigated, and are limited to those supported by the results. In particular, claims of a causative relationship are supported by a well-controlled experimental intervention - partially, and further revisions should eliminate all shortcomings
• Speculation is welcomed, but should be identified as such - yes, there should be no problem
• Replication experiments might be acceptable (provided the rationale for the replication, and how it adds value to the literature, is clearly described); this paper is not considered a ‘pointless’ repetition of well known, widely accepted results - yes, and further aspects have been given with this reviewer's comments (please see below)

Additional comments

Intro
- References are missing, see "(...) and dentin hypersensitivity". Please add a sound reference.
- "(...) enamel white-spot lesions (Markowitz & Carey, 2018; Silva, et al., 2018), (...)." Please add "Youssef et al., 2020", to complete this aspect.
- Please revise for better readability: "The technique is effective in preventing the progression of initial caries in primary and permanent teeth (Faghihian et al., 2019), AND treatment of early dental caries with resin infiltrant achieved excellent clinical results (Lasfargues et al., 2013; Schwendicke et al., 2014; Faghihian et al., 2019; Youssef et al., 2020)." Insert ", and", to complete this sentence.
- Referencing must be improved, see "(Oh HJ et al., 2017)" - delete "HJ".
- The thoughts "In addition, licorice extract has proved to be effective (...). Its action on biofilm limits the drop in pH, (...). The properties of grape seed extract facilitate prevention (...)." do not guide the reader to the main topic of this paper, and would seem misleading. I recommend to summarize these aspects, thereby focussing on your main thoughts on infiltration.
- "We used resin infiltrant to treat sound enamel surfaces, performed artificial demineralization, then measured the changes in surface hardness of the enamel by micro-indentation hardness testing. The demineralization depth and degree of penetration were assessed using confocal laser scanning microscopy (CLSM). Surface morphological changes of the enamel before and after resin infiltration, artificial demineralization, and simulated toothbrushing were observed using scanning electron microscopy (SEM), and the effect of resin infiltrant on resistance of tooth enamel to acid erosion and demineralization was explored." This would seem like a summary of your methodology only. Please shorten considerably remember that these aspects must be given with your Meths section, and this would be a simple repetition. Again, please elaborate your aims more clearly - clarify WHY THIS STUDY WAS NECESSARY, and what (missing) aspects will be added to the literature.
- A clear and reasonable null hypothesis still is missing. Remember that H0 must be deducible from the forgoing thoughts.

Meths
- "The Stomatological Hospital of Chongqing Medical University granted ethical approval to carry out the study within its facilities." Again, please add number and date of approval of your Ethical Vote.
- "(...) using a hard-tissue cutting machine." Again, please add (brand name; manufacturer, city, country) in parentheses.
- Same with your "Vickers microhardness tester", "beaker", "constant-temperature shaker", "incubator", "ImageJ software", "scanning electron microscopy", and so on (this list is not complete, so double check and revise your complete text, please).
- See, for example, "electric toothbrush (Oral-B, Braun, Germany)". This must read "electric toothbrush (please insert model name here; Braun Oral-B/Procter & Gamble, Schwalbach am Taunus, Germany)". Again, this must be followed with ALL materials (including chemicals) and methodologies (including statistical software), so please use general names with your text, followed by (brand name; manufacturer) in parentheses. Stick to semicolon. Revise thoroughly.
- Why should you do this? Remember that reproducibility is the cornerstone of scientific advancement, and note that outputs like exact methodology protocols empower researchers to go one step further in contextualizing their work to ensure it remains replicable.
- Do not use legal terms with your text. Delete "GmbH", "Corporation" (or "Inc.", "Corp.", "™", "®", and so on).
- Again, please revise for sound grammar. "Then, these specimens ultrasonically cleaned (...)" must read "Then, these specimens were ultrasonically cleaned (...)."
- Again, "(IBM SPSS 20.0 for Mac; IBM, United States)" should be perfectible. "(SPSS 20.0 for Mac; IBM Analytics, Armonk, NY, USA)" would seem right.
- As a polite and cordially remark, please note that it is not considered the task of a reviewer to co- or ghost-author your draft. This is exclusively the authors' task, please remember that all authors have "read and approved" this revised manuscript. Astonishingly, this revised and re-submitted draft still suffers from several minor and major shortcomings.

Results
- "As shown in Table 1 (Fig. 1), (...)" - please note that (with your text, your table, and your figure) your results will be tripled. This is not acceptable, please delete Table 1. All important aspects would seem covered with Fig. 1.
- "(P = 0.25), indicating" must read "(P = 0.250), indicating". Please provide exact P values, and give 3 decimal places.
- Consequently, "P = 0.0001" must read "P < 0.001".

Disc
- What about H0? Please remember that your null hypothesis can be rejected or not rejected. Please revise carefully.
- "MH can be used as a parameter to detect changes in the mineral content of dental hard tissue (Gomez et al., 2008); MH changes can reflect mineral loss or gain." Indeed, there is a strong correlation between MH and mineral content. Refer to https://pubmed.ncbi.nlm.nih.gov/10217515/ , and discuss.
- "This result is consistent with findings from studies investigating the effect of etching gel on enamel surfaces conducted by Meyer-Lueckel et al. (Meyer-Lueckel, Paris & Kielbassa, 2007; Paris, Dörfer & Meyer-Lueckel, 2010; Neuhaus et al., 2013; Arnold et al., 2015)." There are several papers cited, so this sentence must read "This result is consistent with findings from previous studies investigating the effect of etching gel on enamel surfaces (Meyer-Lueckel, Paris & Kielbassa, 2007; Paris, Dörfer & Meyer-Lueckel, 2010; Neuhaus et al., 2013; Arnold et al., 2015)."
- A paragraphs elaborating both strengths and limitations of the current paper is missing.

Concl
- With your conclusions, please stick exclusively to your revised aims. Do not simply repeat your results here. Instead, provide sound, reasonable and generalizable extensions of your outcome. The latter must be based on your results, for sure, but this section is called "Conclusions", but NOT "Repetitions".
- "Studies have shown (Costenoble et al., 2016; Gulec & Goymen, 2019) that during orthodontic bonding, bracket bonding performed immediately or shortly following treatment of demineralized enamel with resin infiltrant did not affect the bonding quality of orthodontic brackets." This might be right, and this might be given with the Disc section. However, this is not a conclusion deducible from your results.
- "However, this was an in vitro study, so further studies are needed to confirm the feasibility of preventive using resin infiltrant to protect sound tooth enamel against erosion and demineralization." Again, this is neither considered a conclusion sticking to your aims, nor to your results. Revise carefully.

Refs
- Please revise for uniform formatting of ALL references given.

In total, this revised and re-submitted draft has been considerably improved, and still would seem interesting. However, a further round of revisions would seem mandatory, and this must stick to all the comments given above. Re-review is strongly recommended.

Reviewer 3 ·

Basic reporting

The paper was improved, and the new references were insert. Only need to improve the proposition - line 147-148 please rewritten the purpose - like abstract

Experimental design

This paper is a original research, well wrote and with knowledge.
Only need to inser in text the experimental design, due to this part got very confused.

Validity of the findings

The findings were significative, and provide new information. However need to improve the conclusion
Lines 1077-1079 suggestion - " This provides evidence that application of resin infiltrant on enamel surfaces before orthodontic treatment can be promoted resistance of dental demineralization."

Additional comments

line 147-148 please rewritten the purpose - like abstract
Vickers hardness value should normally be expressed as a number only (e.i. 320 or 320HV) without the units (g/um2)
line 331 - why were the polished specimens re-applied with double-layer of acid-resistant nail varnish?
It is important insert experimental design before Resin infiltration and polishing section, due to very confused in text.
In SEM, you could add the information described in text with the insertion of arrows or another way of marking the figure.

---

## Round 0.3 · accepted · Accept

In line with the re-review from Reviewer 2, this article is now Accepted.

Reviewer 2 ·

Basic reporting

This revised and re-submitted paper would seem satisfying.

Experimental design

This revised and re-submitted paper would seem satisfying.

Validity of the findings

This revised and re-submitted paper would seem satisfying.

Additional comments

This revised and re-submitted paper would seem satisfying, and is considered ready to proceed.

2 minor aspects are in need of revision, and this should be possible with the proofs:
- Please note that "The null hypothesis tested was that resin infiltrant can improve the ability of sound enamel surfaces to resist acid erosion and demineralization and completely penetrate an acid-etched, demineralized enamel area." must read "The null hypothesis tested was that the ability of sound enamel surfaces to resist to acid erosion and demineralization will not differ from resin infiltrated enamel previously acid-etched and demineralized."

- "Accept the null hypothesis." must read "Thus, the null hypothesis was rejected.